# Functional Alteration in the Brain Due to Tumour Invasion in Paediatric Patients: A Systematic Review

**DOI:** 10.3390/cancers15072168

**Published:** 2023-04-06

**Authors:** Nur Shaheera Aidilla Sahrizan, Hanani Abdul Manan, Hamzaini Abdul Hamid, Jafri Malin Abdullah, Noorazrul Yahya

**Affiliations:** 1Department of Radiology, Makmal Pemprosesan Imej Kefungsian (Functional Image Processing Laboratory), University Kebangsaan Malaysia Medical Centre, Kuala Lumpur 56000, Malaysia; nsaidilla96@gmail.com (N.S.A.S.); hamzaini@ppukm.ukm.edu.my (H.A.H.); 2Department of Radiology and Intervency, Hospital Pakar Kanak-Kanak (Children Specialist Hospital), Universiti Kebangsaan Malaysia, Kuala Lumpur 56000, Malaysia; 3Jabatan Neurosains, Pusat Pengajian Sains Perubatan, Jalan Hospital USM, Kampus Kesihatan, Universiti Sains Malaysia, Kota Bharu 16150, Malaysia; brainsciences@gmail.com; 4Brain and Behaviour Cluster, Pusat Pengajian Sains Perubatan, Kampus Kesihatan, Universiti Sains Malaysia, Kota Bharu 16150, Malaysia; 5Department of Neurosciences & Brain Behaviour Cluster, Hospital Universiti Sains Malaysia, Kampus Kesihatan, Universiti Sains Malaysia, Kota Bharu 16150, Malaysia; 6Diagnostic Imaging & Radiotherapy Program, School of Diagnostic & Applied Health Sciences, Faculty of Health Sciences, Universiti Kebangsaan Malaysia, Kuala Lumpur 50300, Malaysia; azrulyahya@ukm.edu.my

**Keywords:** brain tumour, fMRI, cognitive, language and speech, motor, visual impairment

## Abstract

**Simple Summary:**

Brain tumours can result in cognitive, language, motor, and visual deficits in paediatric patients by invading the functional regions of the brain and altering neuronal networks. The specific mechanisms of tumour invasion and the cognitive impact of various treatments remain uncertain. In this systematic review, we discuss the functional alteration in the brain of paediatric patients caused by the tumours in different locations. The present study proposes to understand the changes in the functional connectivity of paediatric patients after tumour invasion using resting-state and task-based fMRI.

**Abstract:**

Working memory, language and speech abilities, motor skills, and visual abilities are often impaired in children with brain tumours. This is because tumours can invade the brain’s functional areas and cause alterations to the neuronal networks. However, it is unclear what the mechanism of tumour invasion is and how various treatments can cause cognitive impairment. Therefore, this study aims to systematically evaluate the effects of tumour invasion on the cognitive, language, motor, and visual abilities of paediatric patients, as well as discuss the alterations and modifications in neuronal networks and anatomy. The electronic database, PubMed, was used to find relevant studies. The studies were systematically reviewed based on the type and location of brain tumours, cognitive assessment, and pre- and post-operative deficits experienced by patients. Sixteen studies were selected based on the inclusion and exclusion criteria following the guidelines from PRISMA. Most studies agree that tumour invasion in the brain causes cognitive dysfunction and alteration in patients. The effects of a tumour on cognition, language, motor, and visual abilities depend on the type of tumour and its location in the brain. The alteration to the neuronal networks is also dependent on the type and location of the tumour. However, the default mode network (DMN) is the most affected network, regardless of the tumour type and location.Furthermore, our findings suggest that different treatment types can also contribute to patients’ cognitive function to improve or deteriorate. Deficits that persisted or were acquired after surgery could result from surgical manipulation or the progression of the tumour’s growth. Meanwhile, recovery from the deficits indicated that the brain has the ability to recover and reorganise itself.

## 1. Introduction

Brain tumours in children are relatively rare, accounting for approximately 20% of all childhood cancers [1]. In Malaysia, brain and nervous system tumours are the second most common childhood cancers, with an incidence rate of 2 per 100,000 children [2]. Although brain tumours are less common than other types of cancer, they can cause significant functional impairments due to their location in the central nervous system [3]. Children with brain tumours are at risk of experiencing cognitive dysfunction, including impairments in thinking, learning, and memory. Such deficits can significantly impact a child’s ability to function at school, at home, and in social situations. Therefore, it is essential to evaluate the effects of tumour invasion on neuronal and network alterations, which can help to understand better the cognitive deficits children may experience.

The size and location of a brain tumour can significantly impact cognitive function, as different areas of the brain are responsible for specific functions. The cognitive deficits that have been observed in children with brain tumours include attention and concentration [4,5], memory impairment [5,6,7], language difficulties [8,9], executive function deficits [10,11], and processing speed deficits [12]. In addition to the direct effects of the tumour and treatment, children with brain tumours may also experience emotional and behavioural problems [4,13], such as anxiety and depression, which can further impact their cognitive function. Furthermore, the severity of cognitive impairment is associated with the tumour characteristics, such as size, histology, and growth rate [14].

Functional magnetic resonance imaging (fMRI) is a non-invasive imaging technique that can measure the functional connectivity of the brain in patients with brain tumours. In healthy individuals, different brain regions work together in coordination, with different areas communicating with one another through neural pathways. In patients with brain tumours, however, the presence of the tumour and the associated inflammation and oedema can disrupt the brain’s normal functioning and affect the functional connectivity between different regions [8,9].

Previous studies have suggested that brain tumours can have a significant impact on the functional connectivity of the brain and that different brain networks are affected in different ways depending on the type and location of the tumour [4,5,8,9,15]. This systematic review aims to evaluate the effects of tumour invasion on the cognitive function of patients. The present review also discusses the alterations and modifications in neurology and anatomy concerning treatments.

## 2. Materials and Methods

### 2.1. Search Strategy and Study Selection

Two independent researchers conducted a systematic search using the National Centre for Biotechnology Information (PubMed) electronic database. The preferred reporting items for systematic reviews and meta-analyses guidelines (PRISMA) [16] and previous studies [17,18,19,20,21,22,23,24,25,26,27,28,29,30] were used as the reporting guidelines (Figure 1). The search was performed to identify studies reporting functional magnetic resonance imaging (fMRI) and brain tumours in paediatric patients below 18 years old. We aimed to assess the research on children with brain tumours who underwent fMRI for brain resection surgery and evaluate the alteration of cognitive functions due to the tumour invasion pre- and post-operatively. In addition, we included studies that compared brain tumour patients with a control group and used resting-state and task-based fMRI as interventions.

The article search was conducted between the earliest record and 1 February 2012. Search terms used included ‘brain tumor or brain tumour or brain cancer or brain neoplasm’ and ‘child or children or paediatrics or minor or junior’ and ‘fMRI or functional MRI or functional magnetic resonance imaging’. We manually checked for associated articles in references and citations through the Google Scholar database. All records were grouped into a final database after removing duplicates, screening the titles and abstracts, and, full-text article screening, as shown in Figure 1.

### 2.2. PICOS and Inclusion and Exclusion Criteria

The PICOS method was used as the guideline for selecting the articles. The eligibility criteria is summarised in Table 1. Original studies in English were reported in peer-reviewed journals, describing research on participants with brain tumours aged <18 years old who underwent resting-state fMRI and task-based fMRI. No limitation was set for sample size, but the year of research publication was restricted to 10 years from the current date. This is because there are not much clinical fMRI research conducted prior. We excluded articles that utilised other imaging modalities, such as computed tomography (CT), positron emission tomography (PET), ultrasound, navigated transcranial magnetic stimulation (nTMS), magnetic resonance imaging (MRI), and magnetic resonance spectroscopy (MRS). Review papers, systematic reviews, case studies, and technical notes were also excluded.

## 3. Results

### 3.1. Data Extraction and Study Design

Sixteen studies met all the criteria for the systematic review. We used an assessment tool from the National Heart, Lung and Blood Institute—Quality Assessment Tool for Observational Cohort and Cross-Sectional Studies—to assess the quality of the included studies. The assessment evaluated these studies as reasonable and fair (Appendix A). This systematic review was registered under the International Prospective Register of Systematic Reviews (PROSPERO)—CRD42023399695. Upon finalisation of the article selection, information, such as the author’s background, year of publication, study type, country of origin, number of patients, the mean age of patients, tumour type, tumour location and sizes, types of fMRI (resting-state and task-based), and cognition assessment were extracted, as shown in Table 2 and Table 3. The tabulated data in Table 4 were assessed to fulfil our primary objective. The information included (1) a pre- and post-operative examination of the afflicted brain region, (2) post-operative behavioural and cognitive alterations, and (3) a summary of the critical results of the selected papers.

### 3.2. Participants

The sample size of the selected studies was between 6 and 61 per study, including control subjects. The total number of patients reviewed was 363, comprising 172 male patients, 122 female patients and 79 control participants. Participants with an age range between 0 and 18 years were included. The studies were conducted in multiple countries, including China, France, Germany, the United States, Italy, Egypt, and Belarus. Five studies compared brain tumour patients to age- and sex-matched healthy controls [4,9,38,40,41]. However, no study conducted separate analyses based on age and gender. We included participants with all types of tumours in various locations of the brain. Although there have been studies of patients presented with tuberous sclerosis, our analysis only included the participants with the relevant diagnosis (brain tumour). Consequently, we included these two studies [37,42].

We also included an epilepsy study, as it was caused by a benign tumour [34]. The tumour location was not exclusive. However, three reports focused entirely on tumours in the posterior fossa [11,40,41] as they are more common in children than adults [11]. Our primary interest in the present study is to evaluate the changes and alterations in the neuronal networks due to tumour invasion and the correlation with the psychology and neurology assessments. Therefore, the area where the tumour is located and its surrounding areas will be considered.

In order to analyse the alterations caused by the tumour, several brain functions and cognitive networks were explored in relation to the tumour and the affected region. The neurological and psychological impairments and changes in brain activity caused by tumour invasion are covered in the following section. The information is also summarised in Table 5.

### 3.3. Resting-State fMRI

Nine studies used resting-state fMRI (rs-fMRI) to evaluate the neuronal changes and alterations resulting from the tumour invasion [4,8,31,32,33,34,35,36,37]. In these studies, the authors discussed the alterations in patients’ working memory [32,33], language [8,31], sensory or motor [31,36], and visual abilities [31,36]. Subsequently, epilepsy and behavioural inhibition that are caused by tumours were also discussed in two papers [4,34]. One reported study included resting-state and task-based fMRI [37].

#### 3.3.1. Working Memory Assessment

Two studies, using the Weschler Intelligence Scale for Children—4th version (WISC-IV) [43] scores, evaluated patients’ neuropsychological assessments. Both papers reported patients with lateral ventricular tumours [32,33]. He et al. (2020) and Zhu et al. (2017) found that patients’ WISC-IV scores in all indices—verbal comprehension, perceptual reasoning, processing speed, and working memory—were considerably worse post-operatively than pre-operatively [32,33]. Six months following surgery, patients who received a frontal transcortical approach experienced a decline in IQ, perceptual reasoning, and processing speed [33]. Patients who underwent the transcallosal procedure experienced a decrease in IQ and working memory. There were no discernible changes in the two approaches’ total IQ before and after the operation [32].

#### 3.3.2. Language and Speech Alterations

Talabaev et al. (2020) reported five patients with tumours in speech areas [31]. Pre-operatively, none of the patients had a speech impairment; however, two patients had mild language and speech disturbance post-operatively. The disturbances were resolved within a month of intervention [31].

Delion et al. (2015) reported patients with tumours at the supratentorial region and found that two patients presented with language disturbance [8]. One had discreet dysarthria without aphasia, and another had subjective word finding, pre-operatively. However, no objective language disorder was identified. In the remaining patients, Delion et al. (2015) reported that some patients showed deficits or had some impairments persisted (moderate dysarthria, partial right brachial and facial palsy, word-finding problems, reduced verbal fluency, and language disturbances) post-operatively [8]. Although patients’ conditions improved after rehabilitation, some patients still had persistent, discreet abnormalities in word finding [8].

#### 3.3.3. Sensory or Motor Alterations

Three studies included sensory or motor changes [31,36,37]. Anwar et al. (2022) reported patients with tumours at the supratentorial region and two out of twenty-two patients had seizures pre-operatively, localised in the motor area [36]. Both patients developed post-operative motor deficits, with one patient developing a deficiency in the right arm and the other in the left arm.

Talabaev et al. (2020) reported that none of the four patients with tumours in the motor cortex had motion dysfunctions pre-operatively [31]. One patient with pilocytic astrocytoma (PA) in the precentral gyrus and extra motor cortex experienced mild distal hemiparesis post-operatively. MRI revealed a small ischemia area along the front edge of the precentral gyrus. However, all symptoms disappeared within a month [31].

Out of twenty patients, Roland et al. (2017) reported two with sensory or motor alterations due to tumour invasion at the left inferior frontal lobe [37]. One of the patients presented with behavioural changes, irritability, and vomiting. A large enhancing dura-based lesion from the pineal region and tracking along the falx cerebri, tentorium cerebelli, and basal cisterns was discovered, causing a mass effect in the brainstem. Obstructive hydrocephalus was caused by the mass effect occluding the cerebral aqueduct. The sensorimotor network (SMN) is localised in the bilateral precentral and postcentral gyrus according to rs-fMRI, pre-operatively [37].

#### 3.3.4. Visual Alterations

Two studies addressed changes in patients’ visual fields or networks due to tumour invasion [31,36]. Anwar et al. (2022) described a patient with anaplastic pleomorphic xanthoastrocytoma (APXA) who presented with pre-operative partial hemianopia. The visual deficit persisted post-operatively [36]. Talabaev et al. (2020) described a patient with a tumour in the visual cortex who experienced white and black spots or light flashes. However, the patient’s condition following the operation was not documented [31].

#### 3.3.5. Rs-fMRI as Pre-Operative Surgical Planning in Hypothalamic Hematoma Induce Epilepsy

Boerwinkle et al. (2018) conducted a study to investigate the outcomes of epilepsy surgery targeting the sub-centimetre-sized rs-fMRI epileptogenic zone (EZ) in hypothalamic hamartoma (HH) [34]. Overall, seizure reduction was higher in the rs-fMRI group, while the Engel Epilepsy Surgery Outcome Scale was significantly higher in the RS group than in the control group [34].

Roland et al. (2017) reported one patient presented with epilepsy [37]. The presenting seizures were limited to the right face and subsequently progressed to involve the right arm and leg. Diagnostic imaging revealed a lesion in the left inferior frontal lobe consistent with a cavernoma. The patient underwent awake surgery, and post-operatively, the patient’s speech remained fluently intact but was noted to have a mild right facial weakness. Two years post-operation, the patient remained seizure-free and without any neurological deficit [37].

#### 3.3.6. Rs-fMRI Evaluations and Brain Activation

##### Behavioural Inhibition

Cheng et al. (2019) reported patients with diffuse intrinsic pontine glioma (DIPG) located in the brainstem [4]. The study discussed patients’ behavioural inhibition that manifested due to tumour invasion. The rs-fMRI results of the 3 groups (17 participants with DIPG; 8 with the deficit in behavioural inhibition, 9 with no deficit in behavioural inhibition, 5 healthy participants as the control group) showed significant differences in terms of the amplitude of low-frequency fluctuation (ALFF) results. The left supramarginal gyrus, left middle frontal gyrus, and right medial superior frontal gyrus showed increased ALFF in patients with deficits in behavioural inhibition. In contrast, other brain regions showed decreased ALFF, including the dorsolateral superior frontal gyrus and right fusiform gyrus [4].

##### Default Mode Network

Zhu et al. (2019) conducted a study on patients with mixed germ cell tumours located at the third ventricle [35]. The regional homogeneity (ReHo), ALFF, fractional ALFF (fALFF) and brain function connections (DMN and Hippocampus as ROI) revealed changes in various regions of the brain at different post-operative time points. Furthermore, the connections of the DMN revealed improved connections in the bilateral middle frontal gyrus and other regions in the first month post-operative, which returned to pre-operative levels three months later. There were no interruptions in the connections between the bilateral hippocampus and related brain regions [35]. 

### 3.4. Task-Based fMRI and Psychological Assessment

Seven studies used task-based fMRI to evaluate brain changes and plasticity due to tumour invasion [9,11,38,39,40,41,42]. In these studies, they discussed the alterations in patients’ working memory [11,40,41], language [9,39], sensory or motor [39], and visual abilities [39,42]. Subsequently, epilepsy and reading interventions that are caused by tumours were also discussed in two papers [38,42].

The task-based fMRI (tb-fMRI) data were correlated with psychological assessments, which included WISC-IV, Weschler Abbreviated Scale of Intelligence (WASI) [44], Child Behaviour Checklist (CBCL) [45], Trains Letter/Number Switching Task and the Delis–Kaplan Executive Function System (D-KEFS) Colour-Word Inhibition/Switching Task [41] and Woodcock–Johnson III battery [38].

#### 3.4.1. Working Memory Assessment

Robinson et al. (2014) found that patients with brain tumours at the posterior fossa had lower cognitive performance than healthy children in all the WISC-IV functional domains [41]. Additionally, patients also had poor working memory capacity on the Trains Letter/Number Switching Task and the Delis–Kaplan Executive Function System (D-KEFS) Colour-Word Inhibition/Switching Task [41]. For patients with brain tumours, overall IQ scores were noticeably lower than those of healthy children. Furthermore, patients with brain tumours showed severe difficulty with concentration on the Child Behaviour Checklist (CBCL), as well as considerable impairments with behavioural regulation and metacognition, and on the general executive composite on the Behaviour Rating Inventory of Executive Function (BRIEF), as reported by their parents [41].

In a study by Hoang et al. (2019), the neuropsychological tests of patients with tumours at the left posterior lobe of the cerebellum revealed that their verbal comprehension index, perceptual reasoning index, and processing speed index scores were significantly lower than those of healthy controls [40]. There was no significant difference between the control group and the patient group’s performance scores on the working memory test that involved the phonological loop. However, patients performed significantly worse than controls on the compound stimulus visual task information (CSVI), which involved the visuospatial sketchpad. In the case of ordered span and transposed span, patients again performed worse than controls [40].

#### 3.4.2. Reading Scores Evaluation

Zou et al. (2016) conducted a study on 42 patients with medulloblastoma in the cerebellum, comparing the reading intervention (INT) and standard-of-care (SOC) on reading abilities [38]. The results showed no significant differences between INT and SOC for any reading scores at Time Windows 0, 1 or 2. Sound awareness was significantly higher for the INT group in Time Window 3. Both patient groups scored significantly lower in fluency compared to controls (TD), with the SOC group scoring significantly lower in word attack and sound awareness. The generalised estimating equation (GEE) model showed no significant differences in longitudinal changes among the groups, but when each Time Window was examined separately with the Wilcoxon rank sum test, there was a declining trend for the reading scores in the patients, except for the sound awareness score in INT [38].

#### 3.4.3. Working Memory

In a study by Hoang et al. (2019), for n-1 back tasks, the results showed no significant differences between patients with brain tumours and controls for accuracy rates in visual-verbal (VIVE), visual non-verbal (VINV), auditory verbal (AUVE), and auditory non-verbal (AUNV) tasks [40]. There was an increase in mean reaction times in patients for VINV only, while no difference was found for VIVE. For n-2 back tasks, results showed a decrease in accuracy rates in patients for VIVE, AUVE, AUNV, and VINV. They also revealed trends for VIVE, VINV and AUNV, as well as increased mean reaction times in patients [40].

In contrast to the reaction times, Robinson et al. (2014) reported that patients and healthy controls performed similarly in all levels of the n-back task [41]. However, there is a significant difference in accuracy; patients’ accuracy on their overall task was significantly worse than healthy controls. Meanwhile, patients’ performance on 0-back and 2-back levels was less accurate than healthy controls [41].

Wolfe et al. (2013) reported that in patients with tumours located at the posterior fossa, as task demand (working memory load) increased, the average reaction times among patients also increased across conditions [11]. A paired sample two-tailed *t*-test disclosed significant changes in reaction times from 0-back to 1-back, and from 0-back to 2-back—as difficulty increased, reaction times grew slower [11].

#### 3.4.4. Tb-fMRI as Pre-Operative Surgical Planning

##### Language

Lorenzen et al. (2018) reported that pre-operative fMRI and/or dMRI tractography results indicated that gross total resection (GTR) resulted in permanent neurological deficits in 17 out of 28 cases [39]. Three patients had their surgical approach changed and one patient had a vigorous activation of the visual word form area close to a left mesial-temporal tumour, replicated over the four language tasks. As a result, a sub-occipital, transtentorial access route was chosen for GTR, and no post-operative language disturbance developed, except for temporary memory problems. In four of the twenty-eight cases, imaging results indicated that GTR would not result in post-operative neurological deficits. The fMRI results confirmed left hemispheric dominance for language in three of the four cases. Eight children had transient post-operative sequelae, including one with conduction aphasia [39].

##### Sensory or Motor

Lorenzen et al. (2018) presented a patient with pleomorphic xanthoastrocytoma of the left precentral gyrus, and the fMRI revealed that the tumour was close to the hand motor area and the cortical spinal tract (CST) [39]. After intraoperative neuropsychological monitoring, it was decided that GTR could not be performed, and thus a few millimetres of tumour remained along the CST. Immediate post-operative radiotherapy was administered, and the patient had been tumour free for 3 years. One of the patient’s surgical approach was altered based on imaging results that showed a typical activation in the hand motor area and an atypical activation close to the tumour. However, intraoperative cortical/subcortical stimulation proved this to be a false positive. One patient with a tumour near the motor cortex developed a mild motor coordination disorder in the post-operative period [39].

##### Visual

Lorenzen et al. (2018) reported neurological deficits in 17 out of 28 cases, leading to changes in surgical approach in 3 out of the 17 cases [39]. In one of three cases, the fMRI revealed a localised visual cortex near an occipital tumour. Post-operatively, four patients developed permanent neurological deficits; three of which were visual deficits. One patient developed a visual field defect where the optic radiation was demonstrated to encompass the intra-ventricular lesion. The other patient developed a small visual field after intervention for an occipital tumour. The third patient had a partial visual defect post-operatively due to a tumour in the left temporal lobe [39].

#### 3.4.5. Tb-fMRI Evaluations and Brain Activation

##### Working Memory

Robinson et al. (2014) conducted a study on patients with tumours at the posterior fossa and found that they demonstrated a greater activation signal in the left dorsal anterior cingulate cortex (DACC) than healthy controls during the n-back task [41]. Patients also showed a significant increase in the BOLD signal in this region during the 1-back and 2-back tasks, while healthy controls showed a significant decrease. Analysis of the main effect of n-back load level showed an increase in the BOLD signal bilaterally in various regions, and participants with greater BOLD signal response in the middle frontal gyrus and left medial frontal gyrus had higher task accuracy and performed better on cognitive tasks. However, participants with greater BOLD signal response in left DACC had worse accuracy scores and executive dysfunction in daily life [41]. Wolfe et al. (2013) documented the group analysis of activation for all working memory conditions [11]. After subtracting out the baseline activation, patterns of frontal-parietal activation were noted across all patients. Specifically, seven clusters of significant activation were pointed out in the right inferior frontal gyrus, supplementary motor area, left insula, right insula, left inferior parietal lobule, right inferior parietal lobule and right middle occipital gyrus [11].

##### Brain Activation to Language Task and Lateralisation Indices

Riva et al. (2019) reported patients with tumours located at the cerebellum and found that activations in the cerebellum were recorded in the right Crus I, Crus II, and lobule VI, as well as the vermis (lobules IV, V, VI, and VII) [9]. In five patients, the left cerebellum was activated. Two patients had a high prevalence of activations in the right cerebellum, and the patient with the most prominent right cerebellar surgical cavity had almost entirely left-sided activations. In all patients, the left inferior frontal gyrus and insula, as well as contralateral regions in the right hemisphere (albeit to a lesser extent and intensity than the left side activations), were activated. The bilateral supplementary motor area (SMA), precentral gyrus, superior and middle frontal gyri, and anterior cingulate gyrus were all activated. For lateralisation indices (LI), the typical pattern of right lateralisation in the cerebellum and left lateralisation in the frontal lobe were seen in two patients. Two more patients had bilateral activation, and in another, left lateralisation in the cerebellum corresponded to right lateralisation in the frontal lobe. The patients’ Lis two-tailed correlation confirmed a trend to the significance that right lateralisation in the cerebellum correlates with left lateralisation in the frontal lobe [9].

## 4. Discussion

The present study evaluated the functional brain, neurological and psychological changes associated with tumour invasions in paediatric patients utilising rs-fMRI and tb-fMRI. We also discuss the alterations and modifications in neurology and anatomy in relation to the treatment, such as variations in surgical techniques. We also discuss the alterations in neuronal networks that involved working memory, languages, speech, and sensory or motor. We also report a few articles that discuss epilepsy and behavioural inhibition due to tumour invasion.

### 4.1. Resting-State fMRI

#### 4.1.1. Alterations and Modifications Due to the Tumour Invasion

##### Visual

Talabaev et al. (2020) and Anwar et al. (2022) suggest that visual deficits would develop or persist post-operatively in patients with brain tumours [31,36]. The same results were reported in several studies. Goldenberg-Cohen et al. (2011) reported that all patients with brain tumour had optic atrophy and vision loss—four of them experienced vision loss in the pre-operative period [46]. A review by Wan et al. (2018) reported that some patients with craniopharyngiomas were found to meet the criteria for legal blindness, and more than half had some degree of visual impairment pre-operatively [47]. Previous studies also suggested that tumours in the occipital lobe could cause homonymous hemianopia, which is higher in the paediatric population than in adults [48]. In normal maturation, the occipital cortex continues to develop until approximately 8 years of age [49]. If normal vision is not restored before the end of this period, there is a potential that vision loss is permanent [49].

In a study by Harbert et al. (2012), children previously diagnosed with a primary brain tumour had unrecognised visual field defects that were later discovered on systematic neuro-ophthalmic examination [50]. The main cause for visual loss in patients includes autoregulatory vascular changes followed by the direct compression caused by the tumour [46]. Blindness can occur if a brain tumour puts enough pressure on the optic nerve. Many patients experience gradual vision loss, beginning with blurry vision, double vision, or an increasing blind spot [51,52,53]. As the tumour grows, it compresses the optic nerve, resulting in more significant vision loss. Wan et al. (2018) also reported that the main cause of vision loss in patients with brain tumours is due to optic nerve pallor, in which the optic nerve axons are damaged [47]. In a study by Peeler (2017), patients with juvenile PA did better in visual acuity than patients with medulloblastoma and ependymoma [54]. This further suggests that permanent vision loss experienced by patients is related to the severity and duration of hydrocephalus and papilledema [54]. The juvenile PA was the most common tumour type causing an unrecognised visual defect. Furthermore, the most frequent locations were the temporal lobe lesions and the hemispheric lesions [50]. This report aligns with the present review that demonstrated patients with tumours located in the supratentorial region [31,36]

##### Language

Delion et al. (2015) and Talabaev et al. (2020) reported variable patterns observed regarding language function post-operatively [8,31]. Patients experienced new or worsening existing language disorders in the post-operative period. However, more than half of the conditions improved or resolved within a month or within 6 months post-operative [8,31]. Patients experienced language or speech impairments because the tumour is in or near areas of the brain that control language and speech, which in this case, is the Broca’s area and the Wernicke’s area. These regions contain motor neurons that utilise speech production. Damage to the Broca’s area can lead to a breakdown between one’s thoughts and one’s language abilities. Thus, patients often feel that they know what they wish to say but are unable to produce the words [55]. Meanwhile, damage to the Wernicke’s area can lead to aphasia, in which patients can speak fluently, but their speech is often nonsensical and lacks meaning [56]. 

Several studies reported that patients with pre-operative aphasia have a higher chance of persisting language impairments post-operatively [8,57,58]. Desmurget et al. (2007) suggested that functional recovery in the post-operative period may involve a combination of neuroplastic mechanisms [59]. Ilmberger et al. (2008) proposed that neuroplastic mechanisms may be less effective in older patients who may have previously relied on these mechanisms before surgery [58]. On the other hand, Delion et al. (2015) noted that the brain plasticity of young children would become similar to adults after the first 5 years of life and limited to local and reduced plasticity in the area close to the tumour [8].

##### Epilepsy

Epilepsy is a common manifestation of primary brain tumours [60]. Michelucci et al. (2013), reported that seizures were the first symptom of a tumour in almost all patients [61]. Patients with brain tumours may have seizures because the cells around the tumour have developed abnormally or the tumour have caused an imbalance of chemicals in the brain. Both of these can interfere with normal electrical activity in the brain [62,63]. Previous studies demonstrated that patients with brain tumours that developed epilepsy in the pre-operative period recovered or became seizure-free post-operatively [64,65,66]. However, Mohan et al. (2008) reported that some patients were only free from recurring seizures for 5 years and some for at 10 years after surgery [66]. Fallah et al. (2015) reported that patients with ganglioma demonstrated the highest probability of long-term seizure freedom, followed by patients with DNETs and gliomas [64]. Patients who were operated on before 15 years old have a lower chance of recurring seizures than those who were operated on after 30 [65]. Similarly, patients with a shorter duration of epilepsy also have a low probability of recurring seizures compared to those with a longer epilepsy duration [65,67]. This indicates that the age and time of epilepsy played a role in the clinical outcome. In addition, temporal location and completeness of resection demonstrated a trend towards longer duration of seizure freedom [64].

##### Hydrocephalus in Tumour Invasion

The mass effect had blocked the cerebral aqueduct of a patient, which resulted in obstructive hydrocephalus [37]. Hydrocephalus can play a significant role in altering the functional network by affecting the structural integrity and function of various regions and neural networks [68,69,70]. The increased pressure in the brain caused by hydrocephalus can compress and damage brain tissue, leading to changes in the structural connectivity between different brain regions [71]. This can result in disruptions of the functional network of the brain, which is the network of interconnected neural circuits that support various cognitive, motor, and sensory functions. A study by Hattori et al. (2011) has shown that hydrocephalus can lead to decreased white matter integrity in regions of the brain and this may cause gait disturbance and dementia [72].

##### Sensory or Motor

A previous study presented patients with acute pure motor hemiparesis due to brain tumours [73]. This may be due to tumour-associated thrombosis or embolism involving one of the deep penetrating arteries [73]. Keilani et al. (2012) reported that patients with glioblastoma experienced deficits in the muscular strength of thigh muscles, and their general physical performance was lower than expected [15]. The deficits persisted in the post-operative period, which may be due to the impact of the clinical course of glioblastoma on patients’ muscular strength [15].

A study by Piscione et al. (2014), which investigated the physical functioning of patients with posterior fossa tumours, found that patients demonstrated decreased physical functioning, most notably in balance [74]. However, the degree to which these deficits are due to the tumour, and the required treatment and its possible discontinuation over time is unknown [74]. Motor deficits present at diagnosis may improve with the addition of corticosteroids or after surgical debulking, but the extent of improvement depends on severity [75,76]. Overall, several factors can contribute to motor deficits. Pre-operative deficits are often related to oedema, while post-operative deficits may result from surgical manipulation or post-operative oedema [77]. The following section further discusses these alterations and modifications caused by the treatment.

#### 4.1.2. Alterations and Modifications Due to Treatments (Surgical Technique or Radiotherapy or Chemotherapy)

##### Sensory or Motor

Anwar et al. (2022) and Talabaev et al. (2020) demonstrated that patients developed new deficits post-operatively, however, some deficiencies can regress within a month [31,36]. Motor pathways affected by the tumour or inflammation caused by treatment may also cause focal motor deficits [75,76,78]. In acquired surgical motor deficits, it may be due to surgical manipulation in motor pathways that create deficits that are slow to resolve or fail to improve [75,76]. McGirt et al. (2009) documented that some patients developed surgically acquired motor deficits, in which more than half of them had permanent deficits while some of them improved or resolved [78].

The somatosensory and motor systems are closely linked and rely on each other to control movement and coordinate sensory information with motor commands [79,80]. Asanuma and Arissian (1984) concluded that the direct sensory input from the thalamus plays an important role in the control of voluntary movements, and the input from the sensory cortex can compensate for the loss of its functions [79]. This is further proven when motor disturbances following ablation of the sensory cortex were recovered within a week, which suggested that the short-term deficit was caused by a shock produced by the lesion [79]. Additionally, Nudo et al. (2000) concluded that sensory deficits resulting from somatosensory cortex lesions are associated with deficits in fine motor skills [80]. If somatosensory feedback is necessary to perform the motor task accurately, motor performance might decline due to a pure sensory deficit [80]. Therefore, the intact somatosensory system is crucial for recovering motor behaviour and improving other body functions [81]. Research on sensory recovery is limited. In the studies on brain stroke [82,83,84], it is suggested that the brain can reorganise to improve somatosensory recovery and the lesion location and course of a stroke may influence the process. This leads to the heterogeneity of brain injury outcomes as recovery differs among individuals.

##### Working Memory

He et al. (2020) and Zhu et al. (2017) used WISC-IV to evaluate patients with lateral ventricular tumours before and after surgery [32,33]. Both studies found that working memory and general intelligence (processing speed, perceptual reasoning, working memory and verbal comprehension) were at an average level in the post-operative period. However, the scores decreased post-operatively. This suggests that the damage to patients’ working memory is caused by the surgery [32,33]. However, the alteration to the neuronal networks and pathophysiological effects because of tumour invasion cannot be ruled out. A few studies reported deficits in patients’ working memory [6,7,85,86,87]. 

Many studies reported the same finding that patients with a brain tumour had significantly reduced cognitive performance compared to healthy controls. The domain reported included IQ (verbal IQ and performance IQ), executive functions, memory, and attention. Hirsch et al. (1979) reported patients with medulloblastoma in the cerebellum [85], while Shen et al. (2013) reported patients with extracerebral tumours and pituitary adenoma [7], and finally, Tucha et al. (2000) reported patients with frontal meningiomas [87]. Furthermore, Margelisch et al. (2015) found that patients with brain tumours performed significantly worse in tests of working memory, verbal memory and attention when compared to patients with leukaemia [5].

The possibility of this working memory and general intelligence deficit might be due to tumours that may damage and invade or disrupt the brain’s neuronal networks. For instance, a tumour in the frontal lobes, which are responsible for executive functions, such as working memory and planning, can impair these abilities [87]. Additionally, the expansion of the tumour may disturb the neural networks, or the imbalance of tumour-related transmitters might affect the limbic system [5]. Subsequently, tumour size also significantly affected intelligence in the intracerebral subgroup [7,87]. 

He et al. (2020) and Zhu et al. (2017) reported that patients with lateral ventricular tumours showed working memory impairment post-operatively [32,33]. He et al. (2020) and Zhu et al. (2017) postulate that the working memory impairment experienced by patients in the post-operative period relies strongly on the damage caused during tumour resection [32,33]. Several studies have reported that different treatment types play a role in cognitive disability in the post-operative period [5,12,86,88]. In a study by Kahalley et al. (2019), it was reported that patients who received the proton craniospinal irradiation treatment scored worse in working memory indices, processing speed indices and full-scale IQ compared to patients who received the proton focal radiotherapy and surgery [12]. In a study by Peterson et al. (2008), it was found that chemotherapy-only treatment administered to patients with acute lymphoblastic leukaemia (ALL) had an adverse effect on intellectual functioning, especially in perceptual reasoning, working memory and processing speeds [88]. Another study also reported that patients with a brain tumour who received chemotherapy had significantly lower scores post-operatively [86]. The deficits in the post-operative period may be related to chemotherapy toxicity [88,89,90]. Higher doses of radiation to the brain is also associated with more severe cognitive declines [89]. 

In the study by Zhu et al. (2017), the frontal transcortical approach was chosen as the surgical treatment, which allows tumour resections in the frontal horn, foramen of Monro, septum pellucidum, and body of the lateral ventricle [33]. However, it is compulsory to pass through cortical structures for this surgical approach. The MiFG is a common PFC route [33]. The low WISC-IV scores are due to the damage caused towards patients’ prefrontal cortex (PFC) during the surgery, which plays a crucial role in human cognitive function and is the basis for the existence of higher cognitive functions [33]. This can be supported by a finding from Postle (2006) who deduced that PFC actively focuses attention on the relevant sensory representation, selects information and performs executive functions that are necessary to control the cognitive processing of the information [91].

He et al. (2020) reported that patients who underwent an anterior transcallosal approach for tumour resection had damage to the body of the corpus callosum, which is believed to be a fibrous connection that plays a crucial role in the interhemispheric transfer of information [32]. One of the most well-known effects of corpus callosum damage is the condition of split-brain syndrome wherein the two hemispheres of the brain are unable to communicate effectively [92]. This can cause various symptoms, including difficulties with language and communication, problems with spatial awareness and perception, and impairments in performing tasks requiring the coordination of information from both hemispheres.

#### 4.1.3. Brain Plasticity and Functional Recovery

##### Alterations Due to Tumour Invasion

Cheng et al. (2019) reported patients with DIPG, with tumours located in the brainstem and found that more than half of the patients had impairment in behavioural inhibition and poor executive functions—the kind of performance is the same as that of Attention Deficit Hyperactive Disorder (ADHD) [4]. Several studies suggested that PFC is the central brain region that is changed in patients with ADHD [93,94,95,96] and that the executive functions deficit is the major deficit in ADHD [97]. Another study also suggested that inhibitory control relies on the PFC and basal ganglia [98]. Additionally, several studies also proposed that executive functions rely on a region in the right frontal gyrus [99,100] and associated subcortical regions, especially for dorsolateral PFC (superior frontal gyrus and MiFG) [4]

ALFF is an rs-fMRI indicator used to detect the regional intensity of spontaneous fluctuations in the BOLD signal, which pinpoints the spontaneous neural activity of specific regions and physiological states of the brain [101]. Neural network abnormalities in an rs-fMRI that correlate with cognition may suggest a cognitive impairment. Cheng et al. (2019) reported that patients with behavioural inhibition showed decreased ALFF in the left dorsolateral superior frontal gyrus and right fusiform gyrus, and increased ALFF in the left supramarginal gyrus, left MiFG, and right medial superior frontal gyrus [4]. The increased and decreased ALFF in those regions suggested a prefrontal hypofunction [4]. Both hypoactivity and hyperactivity can be interpreted as indicating hypofunction in underlying brain structures [102]. Hypoactivity may indicate an inability in a specific area; in other words, it may be considered underpowered. Hyperactivity within a particular area may be regarded as compensatory activity. The brain region may require more energy to perform a simple task. Thus, this condition can be considered inefficient, contributing to patients’ executive dysfunctions [4].

##### Alterations and Modifications Due to Surgery

Working Memory

Zhu et al. (2017) found that the level of co-activation (synchronisation of fMRI signals) in PFC and parietal-occipital brain regions decreased post-operatively [33]. He et al. (2020) reported that the functional connection between the surgical injury area and bilateral cerebral hemispheres decreased significantly but enhanced after 6 months [32]. These findings indicate that the damaged cognitive function can be reshaped [32,33]. This is likely because PFC and corpus callosum are able to reshape via functional compensation, and therefore, cognitive function is preserved. However, the re-emergence of functional activity in areas deprived of neural input and/or deranged neurovascular coupling, whether close or remote to the focal brain injury, is crucial for the recovery of cognitive function [103]. 

Alterations of Default Mode Network

The DMN is a well-established network active at rest and suppressed during tasks requiring attention and decision-making [104]. The DMN typically comprises the posterior cingulate cortex (PCC), praecuneus, inferior parietal, and medial PFC nodes [105]. Most of these selected studies reported disruption in DMN connectivity in patients with brain tumours. The involved area includes the frontal lobe [106,107], temporoparietal lobe [106], and the right and left parietal lobe [108]. Patients had decreased DMN connectivity [106,107,108] and tumour grade and location influenced the level of reduction in DMN connectivity [106,107]. These results are in agreement with several studies on patients with traumatic brain injury [109,110]. 

In the post-operative period, Zhu et al. (2019) reported patients with mixed germ cell tumours located at the brain’s third ventricle [35]. The increase and decrease in ReHo in several brain regions indicated that surgery had effects on the functional activities of some brain regions [35]. Meanwhile, the increase in ALFF and fALFF in the occipital lobe and surgical pathway indicated that residual cerebrospinal fluid in the surgical pathway might interfere with the ALFF value. This is further proven when the values recover to the pre-operative state after a few months [35]. Subsequently, surgery did not interrupt connections of DMN but only caused minimal changes in connection strength. The connections of DMN were restored to a pre-operative state within 2–3 months post-operative [35]. The same result was demonstrated in DMN recovery in traumatic brain injury patients [111,112]. Zhu et al. (2019) also reported that patients experience a variety of post-operative complications, such as recent memory deficit, transient electrolyte disturbance, convulsions, and central hyperthermia which resolve within a certain period of time [35]. The results obtained from the study supported the fact that the functional brain network has plasticity.

### 4.2. Task-Based fMRI

#### 4.2.1. Alterations Due to Tumour Invasion

##### Working Memory

Based on the study by Hoang et al. (2019), patients had significant impairment in intelligence performance and central executive [40]. The same impairments were previously reported in children with medulloblastoma [113,114]. Patients with brain tumours were found to commit more errors in auditory memory than healthy controls [114]. Similarly, patients with posterior cranial fossa astrocytoma and medulloblastoma presented an IQ deficit, attention disorders and procedural memory deficits [113]. Most posterior fossa tumours arise from the cerebellum and surrounding structures [115]. The role of the cerebellum in cognitive function has been examined increasingly in the developmental literature and it is known that the link between the cerebellum and cognition has important implications for children with posterior fossa tumours. Although deficits in intellectual functioning are generally attributed to adjuvant treatment factors, cerebellar damage alone may also contribute to long-term cognitive deficits [115].

Robinson et al. (2014) reported that patients also showed severe difficulty with concentration on CBCL and impairments with behavioural regulation and metacognition in BRIEF [41]. Similar results were reported by Alias et al. (2020), where patients with brain tumours showed statistically significant worse behavioural outcomes than healthy controls in social problems and attention problems [13]. Patients with brain tumours also performed worse in social problems compared to patients with childhood leukaemia [13]. Another study by Fletcher et al. (1990) reported that patients with closed-head injuries had significantly elevated scores on CBCL School and Activities scales compared to patients who had minor or moderately severe injuries [116]. Another study conducted by Loughan et al. (2019) reported that patients with brain tumours presented executive dysfunction in BRIEF, which suggested perceived metacognitive difficulties [10].

Continuous performance tasks are designed to assess sustained attention. Based on the results from Hoang et al. (2019), Robinson et al. (2014) and Wolfe et al. (2012), attention problems in patients may show a similar deficit in activation in regions that subserve attention [11,40,41]. However, these deficits may be amenable to change through cognitive remediation [117].

##### Reading Scores

Based on the study by Zou et al. (2016), the patients’ overall reading scores suggest an apparent preservation of the phonological skills in patients with reading intervention [38]. Similar results were reported by Gillam et al. (2008) and Loeb et al. (2009) in a study of Fast ForWord intervention in children with language impairment [118,119]. This is probably because children with medulloblastoma may have typical neural systems for reading prior to tumour growth. Alternatively, brain networks may have been disrupted by tumour development which in turn respond to intervention differently than typically developing children [38]. However, it is also widely recognised that a condition, known as post-operative cerebellar mutism (PCM), can arise in children following surgery for brain tumours in the posterior fossa. The relationship between PCM and post-operative deficits suffered by medulloblastoma patients is further discussed in the section that follows.

#### 4.2.2. Alterations Due to Surgery

##### Post-Operative Cerebellar Mutism (PCM)

Post-operative cerebellar mutism (PCM) is a well-known complication that can occur in children following surgery for brain tumours in the posterior fossa, including medulloblastoma [120]. PCM is characterised by a temporary or prolonged absence of speech, as well as other neurological symptoms, such as difficulty swallowing, lack of coordination, and emotional and behavioural changes [121]. It is believed that the cause of PCM is related to damage of the cerebellum, which is responsible for motor coordination, balance, and speech.

Patients with medulloblastoma, particularly those who have undergone surgery in the posterior fossa area, may experience a range of neurological deficits that can impact their cognitive and academic functioning. In particular, deficits in speech and language can affect reading abilities, as reading requires the integration of various language-related skills, such as phonological processing, word recognition, and comprehension. These studies had reported the occurrence of PCM in patients following the treatment of medulloblastoma and documented their conditions in the post-operative period [38,40,120,122,123]. It is unclear but deficits experienced by patients in the post-operative period may be due to the damage of the cerebellum which leads to PCM, thus, causing impairments in working memory [40] and language processing, which are critical components of reading ability [38].

#### 4.2.3. Alterations and Modifications in Brain Activation Associated with Tumour Invasion

##### Working Memory

Hoang et al. (2019) reported that patients had no significant activations in the cerebellum for n-1 and n-2 back tasks, whereas healthy controls had activations in several areas [40]. According to the activations in the healthy controls, verbal tasks relate to the right side and non-verbal tasks relate to the left side of the posterior lobe [40]. Similar findings are reported in several functional neuroimaging studies that emphasise the involvement of the left posterior cerebellar lobe in visuospatial working memory [124,125,126]. Hoang et al. (2019) also reported the same finding [40]. Moreover, Hokkanen et al. (2006) found that patients with left cerebellar damage perform visuospatial tasks slower [127]. Gottwald (2004) also proposed that the right cerebellar hemisphere is engaged in verbal working memory while the left cerebellar hemisphere plays a role in non-verbal or visuospatial working memory [128]. Several studies also strengthen this finding [129,130,131].

Robinson et al. (2014) reported that healthy controls demonstrated more significant BOLD signals in MiFG and MeFG during task completion [41]. A study suggested that activation in such regions underlies the maintenance of visuospatial attention, particularly when delays are imposed between stimuli and response [132]. Robinson et al. (2014) also reported that patients demonstrated higher BOLD signal in left DACC than healthy controls in all levels of n-back tasks whereas healthy controls showed relative deactivation [41]. This indicates that, even at less cognitively demanding working memory task levels, patients required increased resources in this region [41]. Similar findings were reported in patients with leukaemia [133]. Interestingly, leukaemia patients were able to successfully and accurately complete the n-back task [133], whereas patients with brain tumours performed significantly less accurately than healthy controls [41]. This suggested that due to the greater impact of brain tumours on neurocognitive function, compensation was unsuccessful at the difficult task levels [41].

Wolfe et al. (2013) reported that patients showed typical activation patterns in response to n-back working memory demands, recruiting mainly frontal, and parietal lobe areas [11]. Owen et al. (2005) also reported the same finding in the study of working memory in healthy adults and youth [132]. This indicates that patients recruit similar areas as typically developing individuals to perform working memory tasks, even though they are at an increased risk for cognitive deficits in working memory [11].

##### Language

Riva et al. (2019) reported that functional lateralisation of the cerebro-cerebellar language system is decreased in patients with early right cerebellar lesions, though still following a typical pattern [9]. They also reported that lesions not located in the areas typically involved in language tasks (Crus I and Crus II) can cause reorganisation between two hemispheres or hemispheric language reinforcement of the original lateralisation. Several studies support this finding [134,135,136,137].

Liegeois (2004) reported that four out of five patients with lesions in or near the Broca’s area did not show interhemispheric language reorganisation, but showed perilesional activation within the damaged left hemisphere [136]. Kadis et al. (2007) confirmed this pattern in children with left injury, revealing an anterior language displacement within the frontal cortex [135]. Raja Beharelle et al. (2010) found that patients with pre- or perinatal left stroke activate the same left inferior frontal regions that support language in normal children [137]. All these indicate that in the case of left injury, language representation often shows interhemispheric reorganisation rather than contralateral shifting [134,138].

##### Reading Intervention

Zou et al. (2016) reported that the activation maps of the INT patients resembled more of those in the healthy controls during the four language tasks, while there was little difference in activation among the three groups during the CPT task [38]. This overall group activation pattern suggested an intervention effect, with the INT patients engaging in more typical brain networks during reading-related tasks [38]. Additionally, increased activation was found in the left superior temporal sulcus and the middle temporal gyrus (STS/MTG), as fMRI tasks required more phonological or semantic processing, and the lateralisation of STS/MTG activation during ORTHOPHONO and story reading was similar between the healthy controls and the INT patients. This suggests that phonological processing during reading may be better preserved or strengthened in the INT patients [38]. Similar results were presented by Gebauer et al. (2012), Shaywitz et al. (2004), and Temple et al. (2003), which found that activation of left STS/MTG regions increased in children with dyslexia after phonologically based reading interventions [139,140,141].

Zou et al. (2016) also reported activations in the caudate, putamen, and thalamus regions in the healthy controls and the INT patients during word reading and rhyming [38]. This indicates improved subcortical and cortical connectivity dynamics in the INT patients during rhyming judgements and implicit word reading [38]. This finding is consistent with the notion that subcortical regions are more activated during procedure learning and in children developing reading skills [142,143].

### 4.3. Crucial Role of Neuroplasticity and Therapy

Brain tumours can have a profound impact on the cognitive and physical functioning of children. The deficits experienced by children with brain tumours can vary depending on several factors, including the location and progression of the tumour, the type of treatment used to manage the tumour, and the child’s age and overall health. One of the most common cognitive deficits experienced by children with brain tumours is memory impairment. This may include difficulty remembering new information or retaining previously learned information. Language impairments, including difficulty with comprehension, expression, and naming, may also occur.

Visual and reading impairments are also possible, and children may have trouble with spatial perception, visual recognition, and processing speed. Motor and sensory impairments, which include coordination, balance, and tactile perception, can also be the result of tumour invasion. Additionally, children who develop epilepsy because of a brain tumour may experience seizures, which can impact their cognitive functioning and overall quality of life.

Neuronal changes that occur after brain tumour resection can vary depending on the extent of the tumour and the location of the resection. When a tumour is removed, the brain undergoes a process of neural reorganisation and rewiring. The brain is able to reorganise itself through a process known as neural plasticity. This means that the remaining healthy neurons can compensate for the lost function by establishing new connections and pathways. However, the degree of neural reorganisation that occurs after surgery depends on the location and the extent of the tumour. If the tumour has caused extensive damage or has spread to other parts of the brain, the brain’s ability to recover may be limited. In these cases, the child may experience long-term cognitive deficits.

The most suitable surgical technique must be selected appropriately by considering the type, size and location of the tumour. The main goal of surgery is to remove as much of the tumour as possible while minimising damage to the surrounding healthy tissues. There are different surgical approaches, such as open craniotomy, endoscopic surgery, and minimally invasive surgery. Treating physicians need to carefully consider all of the factors involved and make a recommendation that provides the best possible outcome for the patients.

### 4.4. Limitations

This review is limited by the small number of studies that satisfied the review criteria. Additionally, there is variability in the quality of the studies, with some having small sample sizes or methodological limitations that could impact the reliability of the results. There are also studies that did not provide a complete record of a certain number of patients, which may confound the results.

## 5. Conclusions

We have evaluated the alterations of cognitive functions in paediatric patients with brain tumours. Brain tumours can negatively impact the cognitive and physical functions in children, based on factors, such as the tumour’s location, treatment, age and health. Memory, language, visual and reading skills, motor and sensory skills, and epilepsy may also be affected.

It is important to note that the cognitive deficits experienced by children with brain tumours can vary widely depending on these various factors. Some children may experience significant improvements in their cognitive functioning following surgery while others may continue to experience cognitive impairments despite treatment. These deficits can have a profound impact on their quality of life, including their academic and social functioning. As such, it is important for healthcare providers to closely monitor the cognitive functioning of children with brain tumours and provide appropriate interventions to support their development and well-being. A comprehensive approach that includes reading intervention and sensory rehabilitation may be necessary for children with these deficits.

## Figures and Tables

**Figure 1 cancers-15-02168-f001:**
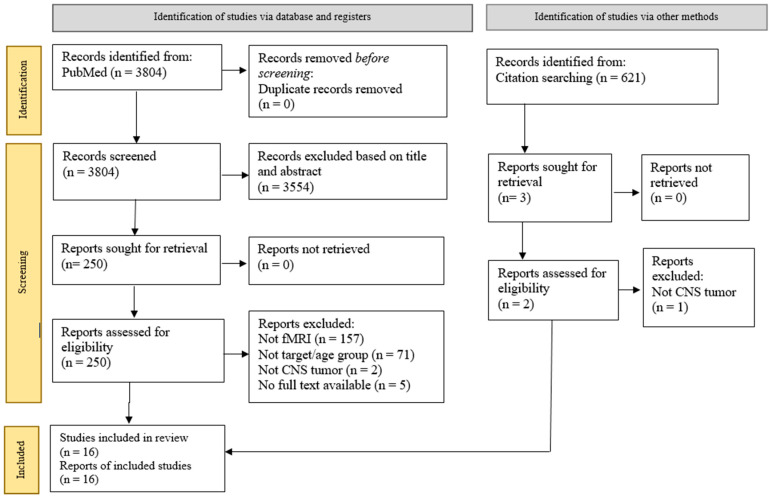
Flow diagram of the PRISMA study selection process.

**Table 1 cancers-15-02168-t001:** PICOS strategy for the selection of study.

PICOS	Criteria
Patient	Children with brain tumours (Age < 18 years old)
Intervention	Resting-state and task-based functional magnetic resonance imaging
Comparison	Types, sizes and location of brain tumours
Outcome	Alteration of cognitive functions/neural network due to brain tumour invasion
Study	All studies related to children with brain tumours examined with functional magnetic resonance imaging

**Table 2 cancers-15-02168-t002:** Demographic data of patients, tumour characteristics and eloquent areas (resting-state fMRI).

No	Author (Year): Country	No. of Patients (Male/Female)	Mean Age (Age Range)	Tumour Type	Tumour Location	Tumour Size	fMRI	RSNs	Eloquent Area	Assessment	Test (Cognition)
1	Talabaev et al. (2020): Belarus [31]	12 (6/6)	14.5 (8–17)	DNET, Ganglioma, PA, GBM	Supra-tentorial region	NR	Rs	NR	Visual, speech and motor cortex	Pre-operative and post-operative	Speech testing
2	He et al. (2020): China [32]	30 (17/13)	9.38 (6–16)	Lateral ventricular tumour	Unilateral ventricle, frontal horn, foramen of Monro, septum pellucidum	21–49 mm	Rs	NR	Working memory	Pre-operative and post-operative	Wechsler Intelligence Scale for Children 4th version: Chinese version
3	Cheng et al. (2019): China [4]	17 (10/7)	7.8 (4–14)	DIPG	Brainstem	NR	Rs	NR	Behavioural inhibition	Pre-operative	Child Behaviour Checklist
4	Delion et al. (2015): France [8]	6 (3/3)	13.6 (11–16)	Cavernoma, ganglioma, ependymoma	Supra-tentorial region	NR	Rs	NR	Language/speech	Pre-operative and post-operative	Object denomination task
5	Zhu et al. (2017): China [33]	15 (8/7)	9.21 (6–14)	Lateral ventricular tumour	Frontal horn, foramen of Monro, septum pellucidum, lateral ventricle	23–47 mm	Rs	NR	Working memory	Pre-operative and post-operative	Wechsler Intelligence Scale for Children 4th version
6	Boerwinkle et al. (2018): USA [34]	36 (24/12)	7.7 (1–18)	Hypothalamic hamartoma	Hypothalamus	0.9–28.3 cm^3^	Rs	NR	Epileptogenic Onset Zone	Pre-operative and post-operative	NR
7	Zhu et al. (2019): China [35]	9 (5/4)	11 (8–14)	Mixed germ cell tumour	Third ventricle	3–5 cm	Rs	Default mode network	Default mode network	Pre-operative and post-operative	NR
8	Anwar et al. (2022): Egypt [36]	22 (15/7)	8.6 (2–18)	Ganglioglioma, DNET, GBM, ETMR, Anaplastic PXA, Anaplastic ependymoma, Meningioma, Anaplastic astrocytoma, LGG, PA, AFH	Supra-tentorial region	NR	Rs	SMN, LAN, DMN, DAN, medial visual and lateral visual network	Motor, language, visual, sensory	Pre-operative and post-operative	Paediatric NIH Stroke Scale
9	Roland et al. (2017): USA [37]	20 (12/8)	12.2 (2–18)	Cavernoma, right cingulate tumour	Left inferior frontal lobe	NR	Rs and Tb	SMN, LAN, DMN, Visual network	Motor, visual, language	Pre-operative and post-operative	Reading paradigm, finger tapping

Abbreviation: DNET: Dysembryoplastic neuroepithelial tumour; PA: Pilocytic astrocytoma; GBM: Glioblastoma; DIPG: Diffuse intrinsic pontine glioma; RS: Resting state; NR: Not recorded; ETMR: Embryonal tumour Multilayered Rosettes; PXA: Pleomorphic Xanthoastrocytoma; LGG: Lower grade glioma; AFH: Angiomatoid fibrous histiocytoma; Tb: Task-based; SMN: Sensorimotor network; LAN: Language network; DMN: Default motor network; DAN: Dorsal attention network.

**Table 3 cancers-15-02168-t003:** Demographic data of patients, tumour characteristics and eloquent areas (task-based fMRI).

No	Author (Year): Country	No. of Patients (Male/Female)	Mean Age (Age Range)	Tumour Type	Tumour Location	Tumour Size	Experimental Design	Task Paradigm	Eloquent Area	Assessment	Test (Cognition)
1	Zou et al. (2016): USA [38]	40 (27/13)	11.7 (Not reported)	Medulloblastoma	Cerebellum	NR	RAN, CPT, implicit reading, ORTHOPHONO, story reading	Event-related	L and R vOT, L and R STS, MTG, L and R IFG	Post-operative	Woodcock–Johnson III Battery, sound awareness, word attack, reading fluency
2	Lorenzen et al. (2018): Germany [39]	26 (13/13)	11.0 (3–17)	LGG, Choroid plexus papilloma	Cortical regions/sub-cortical structures	NR	Beep story, picture story, vowel identification, synonym task, finger tapping, flexion toes	Event-related	Visual, motor, language,	Pre-operative and post-operative	NR
3	Hoang et al. (2019): France [40]	8 (6/2)	13.1 (8–14)	Medulloblastoma	Posterior fossa (left posterior cerebellar lobe)	NR	n-back task (n-1 back and n-2 back); auditory verbal, auditory non-verbal, visual verbal, visual non-verbal	Block design	Working memory	Pre-operative and post-operative	WISC-IV (French), Trail Making Test, D’Alboy Working Memory Procedure
4	Robinson et al. (2014): USA [41]	17 (7/10)	12.6 (8–16)	PA, PFM, DNET, craniopharyngioma	Posterior fossa, parietal lobe, temporal lobe, pituitary gland	NR	n-back task; conditions were presented as 0-back, 1-back, 2-back, 3-back, and repeated in three cycles.	Block design and Event-related	Working memory	Post-operative	CBCL, BRIEF, WISC-IV, D-KEFS
5	Wolfe et al. (2013): USA [11]	9 (8/1)	14.89 (11–18)	Posterior fossa tumour	Posterior fossa	NR	n-back task; 0-back, 1-back and 2-back and counterbalanced in reverse order	Block design	Working memory	Post-operative	Weschler Abbreviated Scale of Intelligence, Lansky Score
6	Riva et al. (2019): Italy [9]	6 (5/1)	11 (8–17)	PA	Cerebellum	5–53 mm	Phonemic verbal fluency task	Block design	Language	Post-operative	WISC-III
7	Li et al. (2013): USA [42]	6 (5/1)	6.3 (2–8)	ATRT, grade 2 astrocytoma	Visual cortex	NR	Alternating black/white screen	Block design	Visual	Pre-operative	NR

Abbreviation: RAN: Rapid Automatized Naming. CPT: Continuous Performance Task. ORTHOPHONO: Orthographic and Phonological processing of letters. L: Left. R: Right. vOT; ventral occipito-temporal area. STS: Superior temporal sulcus. MTG; Middle temporal gyrus. IFG; Inferior frontal gyrus. LGG: Lower grade glioma. NR: Not recorded. WISC: Wechsler Intelligence Scale for Children. PA: Pilocytic astrocytoma, PFM: Plexiform fibromyxoma, DNET: Dysembryoplastic neuroepithelial tumour. CBCL: Child Behaviour Checklist. BRIEF: Behaviour Rating Inventory of Executive Function. D-KEFS: Delis-Kaplan Executive Function System. ATRT: Atypical Teratoid Rhabdoid Tumour.

**Table 4 cancers-15-02168-t004:** Assessment of affected brain area pre-operative and post-Operative, cognitive changes, and its main findings.

No	Author (Year)	Brain Region/Area Affected	Assessment of Affected Brain Region During		Behaviour and Cognitive Changes	Main Findings
			Pre-Operative	Post-Operative		
Resting-state fMRI						
1	Talabaev et al. (2020) [31]	Precentral, postcentral, Wernicke, Broca, visual cortex	Pre-operative fMRI of eloquent brain areas depicted for surgical planning	Eloquent areas were detected during cortex stimulation. No neurological defects post-operative.	Temporary neurological deficit in 3 cases that completely regressed within a month.	ABS towards paediatric PT is quite safe with proper preliminary preparation and planning of each stage of operation.
2	He et al. (2020) [32]	Unilateral ventricle, frontal horn, foramen of Monro, septum pellucidum	WISC-IV scores < 90, prompts below average results (normal level)	All patients undergo FT and AT approach operation scored lower level in all indices of WISC-IV compared to pre-operative scores.	Functional connection between surgical injury area and bilateral cerebral hemispheres ↓ significantly but enhanced after 6 months post-surgery	AT approach impairs patients’ WM and IQ
3	Cheng et al. (2019) [4]	Brainstem	PT showed significant ↓ and ↑ in ALFF in several brain regions	NR	DIPG group with deficit in behavioural inhibition had abnormal brain activities in several brain regions served to deficit in behavioural inhibition processing	Focal spontaneous hyper- and hypofunction findings suggested that PT brain regions may need to exert extra energy to perform task of same degree as HC
4	Delion et al. (2015) [8]	L precentral sulcus, L parietal opercula, L frontopolar area, L parietal postcentral sulcus, L temporal, parietal and occipital lobes	4th PT presented a discreet dysarthria w/o aphasia. 5th PT felt subjective word finding, no objective language disorder identified	Some PT presented with temporary deficits but improved after rehabilitation. Some deficits persisted.	Some PT had discreet abnormalities in terms of memory and language and developed facial palsy, but conditions improved after rehabilitation.	ABS is feasible in paediatric PT and allows GTR w/o major neurologic sequelae
5	Zhu et al. (2017) [33]	Unilateral ventricle, frontal horn, foramen of Monro, septum pellucidum	WISC-IV scores < 100, prompts below average results (normal level)	All PT scored lower level in all indices of WISC-IV compared to pre-operative scores	The level of co-activation and the functional connectivity significantly ↓ after surgery	There is a cognitive dysfunction in paediatric PT with lateral ventricular tumour with 1st year after FT approach
6	Boerwinkle et al. (2018) [34]	Hypothalamus (Epileptic zone)	Pre-operative fMRI of eloquent brain areas depicted for surgical planning	Percentage of total EZ volume ablated was ↑ in 83% of subjects; indicated ↑ accuracy in targeting and ablation of EZ	All PT connectivity patterns demonstrated connectivity to the areas previously reported as the primary areas of ictal propagation	Seizures (gelastic, partial and generalised) were reduced by >30%
7	Zhu et al. (2019) [35]	Third ventricle	ReHo: ↑ mainly in OL, decreased in saddle area ALFF: ↑ value in OL, FL, and cistern around brainstem fALFF: ↑ value in OL and part of FL, ↓ in FL neighbouring frontal horn of lateral ventricle	ReHo: ↑ area was mainly in the OL, ReHo around surgical pathway restored to pre-operative state ALFF: ↑ in OL, FL and cistern around brainstem, areas around surgical pathway gradually ↓ fALFF: Restored to pre-operative state	DMN: Intensity of connection between bilateral FG and other brain regions ↑ in 1-month post-surgery but restored to pre-operative state gradually.	Short-term effects of ReHo, ALFF, and fALFF in brain regions of PT can recover over time.
8	Anwar et al. (2022) [36]	Parietal, temporal, frontal, intra-ventricular, occipital, fronto-parietal	Pre-operative fMRI of eloquent brain areas depicted for surgical planning	SMN was identified in all PT, followed by DMN, and LAN. FC maps of the SMN extracted by SBA were more extensive.	9 patients presented with signs of increased ICT but 8 resolved 3 months post-operative. 10 patients presented with neurological deficits: 1 deceased. 3 months post-operative, improvement in 6 patients, stationary course in 2 patients, 1 showed worsening scale, 1 presented a mass (Ewing Sarcoma)	Rs-fMRI is a promising tool for localising different functional brain networks in pre-operative assessments of brain areas
9	Roland et al. (2017) [37]	Left inferior frontal lobe	Pre-operative fMRI of eloquent brain areas depicted for surgical planning	1st PT: no neurological deficits 2nd PT: full power throughout bilateral upper and lower extremities	1st PT: speech remained fluently intact; had mild right facial weakness. 2 years post-op, remain seizure free. 2nd PT: Suffered a period of aphasia and right hemiparesis. 3 months post-op, completely recovered.	Rs-fMRI is an ideal technology for cerebral mapping in paediatric neurosurgical PT
Task-based fMRI						
10	Zou et al. (2016) [38]	Cerebellum	NR	The activation maps of INT group resembled TD group more than the maps of SOC group during the four language tasks.	Reading scores suggested an apparent preservation of the phonological skills in INT survivors	Sustained neural and behavioural effects of prophylactic intervention in INT survivors
11	Lorenzen et al. (2018) [39]	Cortical regions (visual, motor, language)	Pre-operative fMRI of eloquent brain areas depicted for surgical planning	90% fMRI were interpretable in terms of clearly delineating eloquent cortex	16% PT developed permanent neurological deficits after resection (3 visual, 1 motor)	fMRI plays a profound role for pre-operative risk assessment and decision making, neurosurgical planning, and intra-operative monitoring in PT with LGG
12	Hoang et al. (2019) [40]	Cerebellum (left posterior cerebellar lobe)	PT had lower scores than HC in verbal comprehension, perceptual reasoning and processing speed	PT: no significant activation could be detected in the cerebellum for n-1 and n-2 back tasks. HC: activations related to main effects were mainly found in posterior cerebellar lobe	back: ↑ reaction times in PT for VINV back: ↓ accuracy rates in PT for VIVE, AUVE, and AUNV, ↑ reaction time for AUNV	Cerebellum plays a major role in non-verbal WM.
13	Robinson et al. (2014) [41]	Posterior fossa, parietal lobe, temporal lobe, pituitary gland	NR	PT accuracy ↓ than HC in 1-back, 3-back, and overall accuracy task	PT performed poorly on WISC-IV, D-KEFS and Trails Letter/Number Switching. PT displayed ↑ attention problems and ↓ executive function	PT performed poorly than HC on several cognitive and neurocognitive tasks.
14	Wolfe et al. (2013) [11]	Posterior fossa	NR	Significant activation in RIFG, SMA, LINS, RINS, LIPL, RIPL and RMOG	Average reaction times increase across conditions as WM load increased	Higher cardiorespiratory fitness was associated with better WM across a couple of measures
15	Riva et al. (2019) [9]	Inferior lateral hemispherical zone 1, medial hemispherical zone, inferior vermis, VII A, VIIB, VIIIA-VIIIB, superior vermis, inferior medial hemispherical zone 1, paravermian zone, superior medial hemispherical zone 1, lateral hemispherical zone 2, Crus I-Crus II	NR	Frontal and cerebellar lateralisation of PT is ↓ than HC	PT mean verbal IQ and phonemic fluency scores ↓ than HC	Functional lateralization of cerebro-cerebellar language system is globally ↓ in subjects with early right cerebellar lesion
16	Li et al. (2013) [42]	Visual cortex	Case 2: fMRI revealed right parahippocampal gyrus LGG, visual cortex was adjacent to the tumour	NR	PT visual field remained normal	Sedated fMRI in uncooperative paediatric PT using a passive visual stimulation protocol is safe and reliably predicts functional visual cortex

Abbreviation: fMRI; functional magnetic resonance imaging. ABS; Awake brain surgery. PT; Patients. WISC; Wechsler Intelligence Scale for Children, FT; frontal transcortical, AT; anterior transcallosal, WM; working memory, ALFF: amplitude of low-frequency fluctuation, NR; Not recorded, DIPG; Diffuse intrinsic pontine glioma, HC; Healthy controls, L.; Left, GTR; Gross total resection; EZ; Epileptogenic zone, ReHO; Regional homogeneity, OL; occipital lobe, FL; frontal lobe, fALFF; fractional ALFF, DMN; Default mode network, FG; frontal gyms, SBA; seed-based analysis, ICT; intracranial tension. Rs: resting-state, INT; reading-intervention, TD; typically developing, SOC; standard-of-care, VINV; visual non-verbal, VIVE; visual-verbal, AUVE; auditory verbal, AUNV; auditory non-verbal, D-KEFS; Delis-Kaplan Executive Function System, RIFG; right inferior frontal gyrus, SMA; supplementary motor area. LINS; left insula, RINS; right insula, LIPL; left inferior parietal lobule, RIPL: right inferior parietal lobule, RMOG; right middle occipital gyrus, LGG: lower grade glioma, ↑; increase/higher, ↓; decrease/lower.

**Table 5 cancers-15-02168-t005:** Summary of neurological and psychological deficits and alteration in brain activity due to tumour invasion.

Working Memory		
Authors	Neurological Deficits/Alterations	Brain Activation
Hoang et al. (2019) [40]	WISC-IV: PT < scored compared to HC. No significant difference between PT and HC in the phonological loop. ↑ reaction times in VINV, ↓ accuracy in VIVE, AUVE, and AUNV	No significant activation in the cerebellum for n-1 and n-2 back tasks.
He et al. (2020) [32]	Pre-operative: WISC-IV scores < 90 (normal) Post-operative: PT < scores in all indices of WISC-IV compared to pre-operative	Functional connection between surgical injury area and bilateral cerebral hemispheres ↓ significantly but enhanced after 6 months of follow-up
Robinson et al. (2014) [41]	PT < scores in WISC-IV, D-KEFS, CBCL, and BRIEF compared to HC. PT < accuracy on 0-back, 1-back, 2-back, 3-back, and overall task compared to HC.	n-back task load interactions: PT > BOLD signal in left DACC during all levels in n-back task compared to HC Between-group analyses: HC > BOLD signal in right MiFG and left MeFG
Zhu et al. (2017) [33]	Pre-operative: WISC-IV scores < 100 (normal) Post-operative: PT < scores in all indices of WISC-IV compared to pre-operative	↓ level of co-activation in PFC and parietal-occipital brain regions after surgery.
Wolfe et al. (2013) [11]	NR	Clusters of activation are noted primarily in frontal and parietal areas
Language and Speech		
Riva et al. (2019) [9]	Mean verbal IQ and phonemic fluency scores PT < HC	Frontal and cerebellar lateralization PT < HC
Lorenzen et al. (2018) [39]	One PT showed transient post-operative sequelae (conduction aphasia)	NR
Delion et al. (2015) [8]	4th PT: Presented with discreet dysarthria w/o aphasia 5th PT: Mental clouding that affect language fluidity	NR
Talabaev et al. (2020) [31]	2 PT developed mild disorders post-operation	NR
Motor or Sensory		
Anwar et al. (2022) [36]	2 PT presented with attacks of localised motor seizures w/o deficits	NR
Roland et al. (2017) [37]	Case 4: Mild motor dexterity deficits in left upper extremity, but demonstrates full strength in power	The SMN localised to the bilateral precentral and postcentral gyrus
Lorenzen et al. (2018) [39]	¼ PT with permanent deficits involved sensorimotor	NR
Talabaev et al. (2020) [31]	1 PT had mild distal hemiparesis post-operative	NR
Visual Network		
Lorenzen et al. (2018) [39]	3 PT presented with visual deficits post-operative	NR
Anwar et al. (2022) [36]	1 PT showed partial hemianopia	NR
Talabaev et al. (2020) [31]	PT experienced white and black flashes	NR
Epilepsy		
Li et al. (2013) [42]	PT presented a history of several weeks of complex partial epilepsy	Overlays demonstrated primary visual cortex activation
Roland et al. (2017) [37]	1st PT: bimple partial seizure limited to right face and subsequently involving right arm and leg. 4th PT: began having seizures post-operative and persisted despite treatment with multiple AEDs	1st PT: SMN localised in the precentral and postcentral gyrus laterally, including part of the posterior superior frontal gyrus representing SMA. 4th PT: SMN localised to bilateral precentral and postcentral gyrus.
Boerwinkle et al. (2018) [34]	PT presented intractable epilepsy due to HH	rs-fMRI demonstrated connectivity to the primary areas of ictal propagation
Behaviour		
Cheng et al. (2019) [4]	PT demonstrated deficits in behavioural inhibition	↑ and ↓ ALFF in several brain regions
Default Mode Network		
Zhu et al. (2019) [35]	↑ of ReHo and ALFF values in several brain regions, ↓ fALFF in several brain regions	↑ intensity of connection between bilateral FG and other brain regions
Reading Intervention		
Zou et al. (2016) [38]	Reading scores indicate visible preservation of phonological skills in PT	Line orientation judgement: right hemisphere Letter identification: left hemisphere language areas and sub-cortical areas

Abbreviation: WISC: Wechsler Intelligence Scale for Children, HC: healthy control, PT: patients, VINV: visual non-verbal, VIVE: visual verbal, AUVE: auditory verbal, AUNV: auditory non-verbal, D-KEFS: Delis-Kaplan Executive Function System, CBCL: Child Behaviour Checklist, BRIEF: Behaviour Rating Inventory of Executive Function, BOLD: blood oxygen-level dependent, DACC: dorsal anterior cingulate cortex, MiFG: middle frontal gyrus. MeFG: medial frontal gyrus, PFC: prefrontal cortex, NR: not recorded, SMN; sensorimotor network, AED: anti-epileptic drug, SMA; supplementary motor area. HH; hypothalamic hamartoma ALFF: amplitude of low-frequency fluctuation. ReHo; regional homogeneity. fALFF: fractional ALFF, FG; frontal gyrus, ↑; increase/higher, ↓; decrease/lower.

## Data Availability

Not applicable.

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
