# Peer review of "Functional Alteration in the Brain Due to Tumour Invasion in Paediatric Patients: A Systematic Review"

_cancers, 2023, doi:10.3390/cancers15072168_

Round 1

Reviewer 1 Report

The authors have picked an important and interessting topic and did a lot of good and solid literature research. However, they produced a summary of individual papers (basically the whole section 3 could and probablly should be summarized in one table). 

A review should combine the individual data sets to a new coherent thesis. Conext should be presented in the introduction, limitations should be discussed and conclusions drawn. Which picture ermerges from combining the data? What are the implications of these findings? What needs to be added in terms of information to better understand the subject matter? What new understanding is produced by your review? 

Author Response

Dear reviewer(s),

Thank you very much for the comments and suggestions to improve the manuscript.

Below is the response to the highlighted issues.

Manuscript title: Functional Alteration in the Brain due to Tumour Invasion in Paediatric Patients: A Systematic Review

Item.

Reviewer Comments

Response

1

Reviewer #1

The authors have picked an important and interessting topic and did a lot of good and solid literature research. However, they produced a summary of individual papers (basically the whole section 3 could and probablly should be summarized in one table). 

A review should combine the individual data sets to a new coherent thesis. Conext should be presented in the introduction, limitations should be discussed and conclusions drawn. Which picture ermerges from combining the data? What are the implications of these findings? What needs to be added in terms of information to better understand the subject matter? What new understanding is produced by your review? 

1.     Results presentation and conclusion must be improved.

2.     Provide sufficient background and include all relevant references in the introduction.

Reviewer #2

1.     Improve the introduction and provide sufficient background

2.     Research design should be improved, methods are inadequately described

3.     Results and conclusion must be improved

The authors conducted a systematic evaluation of the effects of tumor invasion on cognition, language, movement, and vision in pediatric patients, by reviewing the previous literature.

While I think this research is very labor intensive and important, I was concerned about the following points.

1. The text was cumbersome and gave the impression of being very difficult to read. Please try using tables and illustrations to make it more comprehensible.

2. Some of the numbers in the headings were incorrect.

Thank you for your constructive review.

The amendment has been done accordingly. Tables summarizing the results have also been added on page 5-15.

Limitations and implications have been added in the conclusion.

Amendment has been done.

Amendment has been done. Table and figure have been added to show to the process and methods used at page 2 and 3.

Tables have been added to summarise the results and findings (page 5-15)

The numbers in the headings have been corrected.

Thank you.

Reviewer 2 Report

The authors conducted a systematic evaluation of the effects of tumor invasion on cognition, language, movement, and vision in pediatric patients, by reviewing the previous literature.

While I think this research is very labor intensive and important, I was concerned about the following points.

1. The text was cumbersome and gave the impression of being very difficult to read. Please try using tables and illustrations to make it more comprehensible.

2. Some of the numbers in the headings were incorrect.

Author Response

(The authors gave the same response as above.)
